# Learning Best Combination for Efficient N:M Sparsity

Yuxin Zhang[1]    Mingbao Lin[2]    Zhihang Lin[1]    Yiting Luo[1]
Ke Li[2]    Fei Chao[1]    Yongjian Wu[2]    Rongrong Ji[1,3,4*]

[1]MAC Lab, School of Informatics, Xiamen University, Xiamen, China
[2]Tencent Youtu Lab, Shanghai, China
[3]Institute of Artificial Intelligence, Xiamen University, Xiamen China
[4]Pengcheng Lab, Shenzhen, China

## Abstract

By forcing at most N out of M consecutive weights to be non-zero, the recent N:M network sparsity has received increasing attention for its two attractive advantages: 1) Promising performance at a high sparsity. 2) Significant speedups on NVIDIA A100 GPUs. Recent studies require an expensive pre-training phase or a heavy dense-gradient computation. In this paper, we show that the N:M learning can be naturally characterized as a combinatorial problem that searches for the best combination candidate within a finite collection. Motivated by this characteristic, we solve N:M sparsity in an efficient divide-and-conquer manner. First, we divide the weight vector into $C_M^N$ combination subsets of a fixed size N. Then, we conquer the combinatorial problem by assigning each combination a learnable score that is jointly optimized with its associate weights. We prove that the introduced scoring mechanism can well model the relative importance between combination subsets. And by gradually removing low-scored subsets, N:M fine-grained sparsity can be efficiently optimized during the normal training phase. Comprehensive experiments demonstrate that our learning best combination (LBC) performs consistently better than off-the-shelf N:M sparsity methods across various networks. Our project is released at https://github.com/zyxxmu/LBC.

## 1   Introduction

Recent years have witnessed the popularity of convolutional neural networks (CNNs) in visual problems such as image classification [34, 12], object detection [11, 8] and semantic segmentation [9, 3]. However, the remarkable performance comes at the cost of huge computation burdens and memory footprint, resulting in great challenges for deploying CNNs in resource-limited devices. Model compression has received extensive attention from both academia and industries. By reducing the number of parameters in CNNs, network sparsity has become one of the most representative techniques to alleviate storage and computation burdens. According to the sparsity granularity, traditional methods can be divided into unstructured sparsity [18, 6, 10] and structured sparsity [20, 29, 27]. As shown in Fig. 1(a), unstructured sparsity removes arbitrary weights in CNNs. Dozens of earlier studies [23, 17, 7] have demonstrated its ability to reach negligible performance degradation under a large compression rate. Unfortunately, unstructured sparsity often results in an irregular sparse matrix, which causes heavy index storage and very little acceleration. On the contrary, structured sparsity removes all weights in a filter as illustrated in Fig. 1 (b). As a result, the sparse weights are still in a hardware-friendly format, leading to notable speedups. Nevertheless, the removal of the whole filters also severely damages the accuracy performance. Thus, future progress is expected to find out a new sparsity pattern to tradeoff the performance and acceleration.

---

*Corresponding author: rrji@xmu.edu.cn

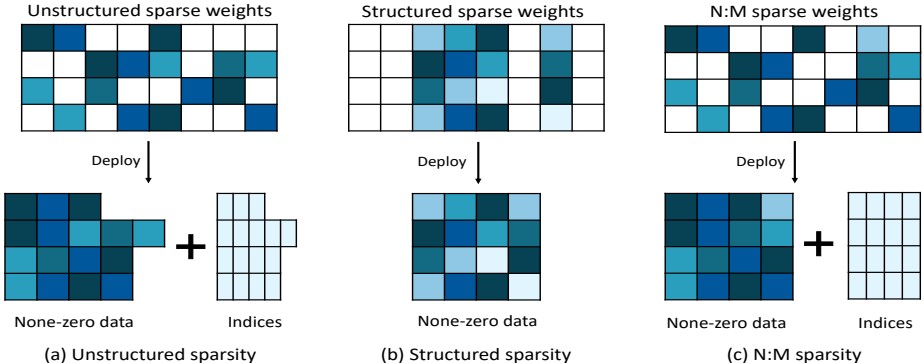

Figure 1: Mainstream sparsity granularity in the literature. (a) Unstructured sparsity removes individual weights at arbitrary locations. (b) Structured sparsity removes entire convolutional filters. (c) N:M sparsity (2:4 case) requires at most N out of M consecutive weights to be non-zero.

More recently, the N:M fine-grained sparsity has shown a promising power for the research community. As depicted in Fig. 1 (c), it divides the weights into several blocks of size M. By preserving at most N out of M weights in each block, it not only achieves a high sparsity rate but also retains a regular sparse matrix structure for an effective acceleration supported by the NVIDIA Ampere Sparse Tensor Core [33]. The pioneering ASP (apex's automatic sparsity) [30] preserves two largest-magnitude weights in every four-element block and then re-trains the sparse network. However, the time-consuming re-training prohibits the deployment of N:M sparsity in resource-constrained scenarios. To avert the pre-training burden, Zhou *et al.* [40] learned N:M sparsity from scratch by leveraging the straight-through estimator (STE) [1] to approximate the gradients of removed weights. Although an additional regularization is proposed to alleviate the weight error accumulation, dense gradients are required to be computed in each training iteration to update both the removed and preserved weights. Consequently, a heavy training burden is unavoidable, barricading the generalization of N:M sparsity to gradient and activation [2]. Thus, how to realize an efficient N:M sparsity remains unexplored.

In this paper, we propose to learn the best combination in order to reach an efficient N:M sparsity (LBC). Our LBC is motivated by the fact that the N:M learning can be modeled as a combinatorial problem which consists in finding, among a finite collection of combinations, one that satisfies all the given conditions. Considering a single block, the purpose of N:M sparsity is to select N non-repeated elements as a combination from the M continuous weights. Thus, N:M sparsity can be simplified as finding the best combination candidate from a total of $C_M^N$ combinations on condition that the training loss can be minimized on the observed dataset. Given this analysis, our LBC solves the N:M sparsity from the perspective of combinatorial problems in an efficient divide-and conquer manner. We first divide the weight block into a collection of $C_M^N$ combinations with a fixed size N. Then, our LBC aims to learn the best combination from this collection. This is achieved by a learnable score variable, which, as we prove in Sec. 3.4, can well tell the relative importance between different combinations. As results, the best combination can be located by gradually removing the low-scored combinations during a normal training process without the necessity of a dense-gradient computation.

We perform extensive experiments to demonstrate the efficacy of our LBC. For instance, our LBC achieves 77.2% top-1 accuracy for training a 2:4 sparse ResNet-50 on ImageNet, which takes the leading position in comparison with existing N:M sparsity methods [30, 40]. It is also worth mentioning that our LBC even surpasses some unstructured state-of-the-art methods that fail to achieve speedups. For example, under a sparsity rate of around 95%, our LBC still achieves a top-1 accuracy of 71.8% with a 1:16 sparse pattern, surpassing the recent competitor STR [17] by 1.2%.

## 2 Related Work

### 2.1 Network Sparsity

Network sparsity has become an active research topic ever since the last decade [18, 10, 25]. Traditional methods can be roughly categorized into unstructured sparsity and structured sparsity. The former category removes individual weights at any position of the network. Prevailing studies remove

unimportant weights using heuristic criteria such as gradient [19], momentum [6], and magnitude [10]. Recent advances implement unstructured sparsity in a training adaptive manner. RigL [7] alternately removes and revives weights according to their magnitudes and dense gradients. Ding *et al.* [6] gradually zeroed out the redundant weights by categorizing CNN weights into two parts at each training iteration and updating them using different rules. Despite the progress, the irregular sparse weight tensors rarely support speedups on common hardware unless the sparse rate increases to 95% or even higher [37, 33]. On the other hand, structured sparsity gains noticeable speedups thanks to its coarse-grained removal of the entire filters. Two issues are widely discussed in existing studies including the pruned network structure and the filter importance measurement. Typical works solve the former issue by rule-of-thumb designs [13, 5] or layer-wise sparse ratio searching based on evolutionary algorithms [21, 24]. As for the second issue, most works resort to devising a certain criterion to estimate the filter importance including output activation [39], ranks of feature map [20], geometric median of filters [14], *etc*.

## 2.2 N:M Sparsity

The N:M fine-grained sparsity [30, 40, 35] advocates N-out-of-M non-zero sparse tensors in the input channel dimension. Supported by the NVIDIA Ampere Core [33], N:M sparsity leads to attractive storage and computation efficiency. For example, N:M sparsity can achieve $2\times$ speedups on the NVIDIA A100 GPU in the 2:4 case, while unstructured sparsity even decreases the inference speed under a similar sparse rate. NVIDIA [30] follows a traditional three-step pipeline to implement N:M sparsity, unfolded as pre-training, pruning and fine-tuning. Pool *et al.* [32] further leveraged channel permutations to maximize the accuracy of N:M sparse networks. Sun *et al.* [35] proposed a layerwise fine-grained N:M scheme to achieve higher accuracy than the uniform N:M sparsity. Recently, there are also efforts in exploring the training efficiency of N:M fine-grained sparsity. Hubara *et al.* [15] devised transposable masks to accelerate the backward phase of N:M sparsity training. Chmiel *et al.* [2] further extended N:M sparsity to activation, which further boosts the training efficiency. Despite the well-retained performance [30], the expensive pre-training process still hinders the deployment of N:M sparsity. Sparse-refined straight-through estimator (SR-STE) [40] learns the N:M sparsity from scratch. Detailedly, the removed weights are also revived during the backward phase using the straight-through estimator [1] and a specially-designed sparse penalty item. Therefore, by choosing N-out-of-M weights with the largest magnitudes in each forward propagation, the N:M sparsity can be obtained in a dynamic and end-to-end training manner. However, it requires dense gradients in order to revive the weights, thus can not be combined with these studies on training efficiency [15, 2].

## 3 Methodology

### 3.1 Preliminaries

Denoting the parameters of an L-layer convolutional network as $\mathbf{W} = \{\mathbf{W}^0, \mathbf{W}^1, ..., \mathbf{W}^L\}$ where $\mathbf{W}^l$ is a $\mathrm{K}^l$-dimension vector[2], N:M fine-grained sparsity divides parameters in each layer into groups along the input channel dimension, leading to each group containing M consecutive weights. It saves computation and retains performance by requiring at most N out of M consecutive parameters to be non-zero values. To this end, the $\mathbf{W}^l$ is firstly divided into $\mathrm{G}^l = \frac{\mathrm{K}^l}{\mathrm{M}}$ groups. For ease of the following representation, we reorganize $\mathbf{W}^l$ as $\mathbf{W}^l \in \mathbb{R}^{\mathrm{G}^l \times \mathrm{M}}$. And then, the optimization of N:M sparsity can be formulated as:

$$\min_{\mathbf{W}} \mathcal{L}(\mathbf{W}; \mathcal{D}) \quad s.t. \quad \|\mathbf{W}_{g,:}^l\|_0 = \mathrm{N}, \tag{1}$$

where $l = 1, 2, ..., \mathrm{L}$ and $g = 1, 2, ..., \mathrm{G}^l$. The $\mathcal{L}(\cdot)$ denotes training loss function and $\mathcal{D}$ represents the observed dataset. An illustrative case of 2:4 sparsity is given in Fig. 1(c).

Previous N:M methods [30, 40] follow the traditional unstructured sparsity [10] to remove weights in a global manner by looking at the weight magnitude. However, they all require huge training costs either from a pre-training phase [30] or dense-gradient calculation [40] to select the most important weights from the large number of parameters in deep networks. Hence, how to achieve N:M sparsity in a more efficient manner remains an open issue so far.

---

[2]Batch norm layers and bias items are omitted here for simplicity.

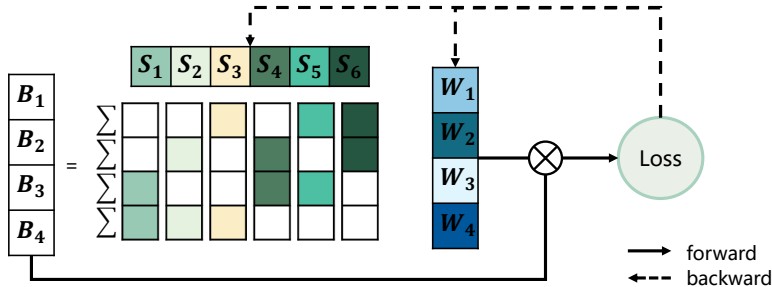

Figure 2: The training process for learning the best combination candidate of N:M sparsity (2:4 case).

## 3.2 Combinatorial Problem in N:M Sparsity

Fortunately, by making full use of the intrinsic particularity in N:M sparsity, the solution space can be well reduced to pursue an efficient N:M sparsity. Concretely, given a weight group $\mathbf{W}_{g,:}^l$, the target of N:M sparsity is to select N out of the M weights in $\mathbf{W}_{g,:}^l \in \mathbb{R}^M$ and discard the remaining M−N ones. This leads to a combination collection $\Theta_g^l = \{\Theta_{g,i}^l\}_{i=1}^{C_M^N}$ where each combination $\Theta_{g,i}^l$ is a subset of $\mathbf{W}_{g,:}^l$ and consists of N non-repeated elements from $\mathbf{W}_{g,:}^l$. It is intuitive to derive the collection size $|\Theta_g^l| = C_M^N$. As a toy example, we have $C_4^2 = 6$ considering the 2:4 case. Also, we have:

$$\Theta_g^l = \Big\{ \underbrace{\{\mathbf{W}_{g,0}^l, \mathbf{W}_{g,1}^l\}}_{\Theta_{g,1}^l}, \underbrace{\{\{\mathbf{W}_{g,0}^l, \mathbf{W}_{g,2}^l\}}_{\Theta_{g,2}^l}, \underbrace{\{\mathbf{W}_{g,0}^l, \mathbf{W}_{g,3}^l\}}_{\Theta_{g,3}^l}, \underbrace{\{\mathbf{W}_{g,1}^l, \mathbf{W}_{g,2}^l\}}_{\Theta_{g,4}^l}, \underbrace{\{\mathbf{W}_{g,1}^l, \mathbf{W}_{g,3}^l\}}_{\Theta_{g,5}^l}, \underbrace{\{\mathbf{W}_{g,2}^l, \mathbf{W}_{g,3}^l\}}_{\Theta_{g,6}^l} \Big\}. \tag{2}$$

According to the above analyses, we realize that the N:M sparsity can be naturally characterized as a *combinatorial problem*, which aims to, given a finite collection of combinations $\Theta_g^l$, search for the best candidate within this collection. Herein, we reformulate the realization of N:M sparsity from the perspective of a combinatorial problem as:

$$\underset{\Theta_{g,i} \in \mathbf{W}_{g,:}^l}{\arg\min} \ \mathcal{L}(\mathbf{W}_{g,:}^l; \mathcal{D}), \ s.t. \ |\Theta_{g,i}| = \mathrm{N} \tag{3}$$

where $l = 1, 2, ..., \mathrm{L}$, $g = 1, 2, ..., \mathrm{G}^l$ and $i = 1, 2, ..., C_M^N$.

**Discussions**. Eq. (1) implements N:M sparsity by globally sparsifying a weight vector $W_{g,:}^l$. In contrast to earlier implementations, our redefined Eq. (3) solves N:M sparsity in a divide-and-conquer manner. We first divide the weight vector into $C_M^N$ subsets of a fixed size N. Then, we conquer the combinatorial problem by scoring each subset and finally picking up the highest-scored one, details of which are given in the following contents.

## 3.3 Learning Best Combination

Based on the above analyses, we further propose to learn the best combination for each weight vector $\mathbf{W}_{g,:}^l$. Following [40], we consider learning the N:M sparsity from scratch to avoid the heavy dependency on an expensive pre-trained model. A learning example of the 2:4 case is shown in Fig. 2. Specifically, denoting the input feature map of the $l$-th convolution layer as $\mathbf{Z}^{l-1}$, the output feature map $\mathbf{Z}^l$ is calculated as:

$$\mathbf{Z}^l = \mathbf{Z}^{l-1} \otimes \mathbf{W}^l, \tag{4}$$

where $\otimes$ denotes the convolutional operation. The earlier study [40] chooses to zero out low-magnitude N weights for every weight vector $\mathbf{W}_{g,:}^l$ in the forward propagation. The straight-through estimator (STE) is used to dynamically remove and revive weights, which however leads to error accumulation on the weights. Besides, dense-gradient computation is of heavy cost since both removed and preserved weights are required to update. Given the combinatorial property, we are entitled to the opportunity to reach N:M sparsity by gradually removing the low-scored candidate subsets in the combination collection $\Theta_g^l$ during the training procedure until the one with the smallest loss can be found. The gradual sparsity is demonstrated to perform better in many traditional network

sparsity [42, 41, 23]. To that effect, we consider the gradual pruning schedule [42] to remove candidate subsets considered unimportant, and the remaining collection at the $t$-th training epoch is:

$$\Theta_{g,t}^l = \Theta_{g,t-1}^l - \bar{\Theta}_{g,t}^l, \tag{5}$$

where $\bar{\Theta}_{g,t}^l \in \Theta_{g,t-1}^l$ contains unimportant candidates at the $t$-th training epoch and its size is:

$$|\bar{\Theta}_{g,t}^l| = \begin{cases} 0, & \text{if } t = t_i, \\ \left\lceil (C_{\text{M}}^{\text{N}} - 1)\left(1 - (1 - \frac{t-t_i}{t_f-t_i})^3\right)\right\rceil - |\bar{\Theta}_{g,t-1}^l|, & \text{otherwise}, \end{cases} \tag{6}$$

where $t \in [t_i, t_f]$, $t_i$ and $t_f$ denote the beginning and ending of the training epochs in the gradual pruning schedule [42]. When $t = t_f$, we remove a total of $C_{\text{M}}^{\text{N}} - 1$ candidate subsets. In other words, only one combination, which is considered to be the best candidate, is preserved in the end.

To measure the candidate's importance, we invent a learnable score matrix $\mathbf{S}^l \in \mathbb{R}^{\text{G}^l \times C_{\text{M}}^{\text{N}}}$. In our design, each element $\mathbf{S}_{g,i}^l$ reflects relative importance of candidate $\Theta_{g,i}^l$ in comparison to others in $\Theta_g^l$ and we reduce the size of collection $\Theta_g^l$ by removing the low-scored candidates. In the forward propagation, an individual weight $\mathbf{W}_{g,i}^l$ often exists in multiple candidate subsets as shown in Eq. (2). We impose a binary mask $\mathbf{B}^l \in \{0, 1\}^{\text{G}^l \times \text{M}}$ upon the weight $\mathbf{W}^l$ and the output in Eq. (4) becomes:

$$\mathbf{Z}^l = \mathbf{Z}^{l-1} \otimes (\mathbf{B}^l \odot \mathbf{W}^l), \tag{7}$$

where $\odot$ represents the element-wise multiplication. The zero elements in $\mathbf{B}^l$ indicate the removal of corresponding weights that contribute least to the network. The binary mask $\mathbf{B}^l$ is derived as:

$$\mathbf{B}_{g,i}^l = \mathbb{I}\left(\left(\sum_j \mathbf{S}_{g,j}^l \cdot \mathbb{I}(\mathbf{W}_{g,i}^l \in \Theta_{g,j}^l)\right) \neq 0\right), \tag{8}$$

where $\mathbb{I}(\cdot)$ returns 1 if its input is true, and 0 otherwise. That is to say, we choose to preserve weights within any preserved candidate subset. Note that if $\mathbf{W}_{g,i}^l$ does not exist in any preserved candidate subset, it will be automatically removed since $\mathbf{B}_{g,i}^l = 0$ according to Eq. (8).

The use of a binary mask originated from many traditional unstructured sparsity methods [7, 16]. However, to our best knowledge, it is the first time to be introduced in N:M sparsity. Moreover, very different from these studies that directly optimize the binary mask, our mask is obtained according to the candidate subsets the corresponding weight falls into. Finally, since the derivative of the $\mathbb{I}(\cdot)$ function is zero almost everywhere, we adopt the STE [1] to update $\mathbf{S}_{g,j}^l$ as:

$$\frac{\partial \mathcal{L}}{\partial \mathbf{S}_{g,j}^l} = \frac{\partial \mathcal{L}}{\partial \mathbf{B}_{g,i}^l}\frac{\partial \mathbf{B}_{g,i}^l}{\partial \mathbf{S}_{g,j}^l} \approx \frac{\partial \mathcal{L}}{\partial \mathbf{B}_{g,i}^l}. \tag{9}$$

Consequently, the STE approximation of weight gradient [40] becomes the approximation of the gradient of the introduced score matrix, avoiding the error accumulation on the weights. Intuitively, removing $\mathbf{S}$ does not influence the result of Eq. (8) in the forward propagation. However, our motive of retaining $\mathbf{S}$ is actually to derive the gradient in the backward propagation, *i.e.*, Eq. (9) to learn the importance of each combination. In Sec. 3.4, we formally prove that the learned score $\mathbf{S}$ can well reflect the relative importance between different candidate subsets. Further, we conduct experiments in Sec. 4.4 to show that the learned score $\mathbf{S}$ well outperforms existing pruning criteria [10, 28].

Our learning the best candidate, dubbed as LBC, is outlined in Alg. 1. Different from traditional methods [39, 26] that conduct network sparsity in a two-step manner including a mask learning and a weight tuning, LBC enables the optimization of $\mathbf{W}$ and $\mathbf{S}$ in a joint framework, leading to a well-optimized N:M sparsity during a normal training phase. It is worthy to highlight the efficiency of LBC since it casts off the dense gradient calculation in [40]. Also, LBC can cooperate with techniques that extend N:M sparsity to gradient and activation to further boost the training efficiency [2, 15].

### 3.4 Tractability of Optimization

We take the candidate $(\mathbf{W}_{g,0}^l, \mathbf{W}_{g,1}^l)$ in 2:4 sparsity of Eq. (2) to show that our introduced mask score can well reflect the relative importance of each candidate subset. The analysis can be applied to other

**Algorithm 1:** Learning Best Candidate (LBC).

---

**Require :** Observed data $\mathcal{D}$; Randomly initialized weights $\mathbf{W}$; Loss function $\mathcal{L}$; Training epochs T; Initial and final epoch for removing combinations $t_i, t_f$; Candidate collection $\Theta$.

**Output :** Trained weights $\mathbf{W}$; Binary mask $\mathbf{B}$.

**1** $\mathbf{S}_{g,i}^l \leftarrow 1\ \forall l, \forall g, \forall i$; // Learnable Importance Score

**2** **for** $t \in [1, \ldots, T]$ **do**

**3**      **if** $|\Theta_{g,t}^l| \neq 1$ **then**

**4**          Calculate $|\bar{\Theta}_{g,t}^l|$ via Eq. (6);

**5**          **for** $l \in [1, \ldots, L]$ **do**

**6**              **for** $g \in [1, \ldots, G^l]$ **do**

**7**                  $\bar{\Theta}_{g,t}^l = \text{Top-}|\bar{\Theta}_{g,t}^l|$ smallest $\mathbf{S}_g^l$ in $\Theta_{g,t-1}^l$;

**8**                  $\Theta_{g,t}^l = \Theta_{g,t-1}^l - \bar{\Theta}_{g,t}^l$; // Remove Unimportant Candidate Subsets

**9**              **end**

**10**          **end**

**11**          Get $\mathbf{B}$ via Eq. (8);

**12**      **end**

**13**      Forward via Eq. (7);

**14**      Update via the SGD optimizer.

**15** **end**

---

N:M patterns as well. The gradient of $\mathbf{S}_{g,1}^l$ is derived using STE [1] as:

$$
\begin{aligned}
\frac{\partial \mathcal{L}}{\partial \mathbf{S}_{g,1}^l} &= \frac{\partial \mathcal{L}}{\partial \mathbf{B}_{g,0}^l} \frac{\partial \mathbf{B}_{g,0}^l}{\partial \mathbf{S}_{g,1}^l} + \frac{\partial \mathcal{L}}{\partial \mathbf{B}_{g,1}^l} \frac{\partial \mathbf{B}_{g,1}^l}{\partial \mathbf{S}_{g,1}^l} \\
&= \mathbf{W}_{g,0}^l \frac{\partial \mathcal{L}}{\partial \mathbf{W}_{g,0}^l} \frac{\partial \mathbf{B}_{g,0}^l}{\partial \mathbf{S}_{g,1}^l} + \mathbf{W}_{g,1}^l \frac{\partial \mathcal{L}}{\partial \mathbf{W}_{g,1}^l} \frac{\partial \mathbf{B}_{g,1}^l}{\partial \mathbf{S}_{g,1}^l} \\
&\approx \mathbf{W}_{g,0}^l \frac{\partial \mathcal{L}}{\partial \mathbf{W}_{g,0}^l} + \mathbf{W}_{g,1}^l \frac{\partial \mathcal{L}}{\partial \mathbf{W}_{g,1}^l},
\end{aligned}
\tag{10}
$$

where $\mathbf{W}_{g,0}^l \frac{\partial \mathcal{L}}{\partial \mathbf{W}_{g,0}^l}$ meets the criterion [28] that estimates the loss changes after removing $\mathbf{W}_{g,0}^l$ as:

$$
\begin{aligned}
\Delta \mathcal{L}(\mathbf{W}_{g,0}^l) &= \mathcal{L}(\mathbf{W}_{g,0}^l = 0) - \mathcal{L}(\mathbf{W}_{g,0}^l) \\
&\approx \mathcal{L}(\mathbf{W}_{g,0}^l) - \mathbf{W}_{g,0}^l \frac{\partial \mathcal{L}}{\partial(\mathbf{W}_{g,0}^l = 0)} + R_1(\mathbf{W}_{g,0}^l = 0) - \mathcal{L}(\mathbf{W}_{g,0}^l) \\
&\approx -\mathbf{W}_{g,0}^l \frac{\partial \mathcal{L}}{\mathbf{W}_{g,0}^l},
\end{aligned}
\tag{11}
$$

which leads Eq. (10) to $\frac{\partial \mathcal{L}}{\partial \mathbf{S}_{g,1}^l} = -(\Delta \mathcal{L}(\mathbf{W}_{g,0}^l) + \Delta \mathcal{L}(\mathbf{W}_{g,1}^l))$. The updating rule of $\mathbf{S}_{g,j}^l$ using stochastic gradient descent (SGD) becomes:

$$
\mathbf{S}_{g,1}^l = \mathbf{S}_{g,1}^l + (\Delta \mathcal{L}(\mathbf{W}_{g,0}^l) + \Delta \mathcal{L}(\mathbf{W}_{g,1}^l)),
\tag{12}
$$

where learning rate and momentum are ignored here for simplicity. The candidate score $\mathbf{S}_{g,1}^l$ is the accumulation of loss change for removing weights in the corresponding candidate. A small score denotes less loss increase thus the corresponding candidate can be safely removed. Thus, our introduced score matrix can be used to measure the relative importance between candidate subsets, and preserving the highest-scored combination well solves the combinatorial problem of Eq. (3).

## 4 Experiments

### 4.1 Experiment Settings and Implementation Details

Our proposed LBC is evaluated on three computer vision tasks including image classification, object detection, and instance segmentation. We implement LBC using the PyTorch [31] upon 2 NVIDIA

Table 1: Results of different methods for training the N:M sparse ResNet-18, ResNet-50 and DeiT-small on ImageNet. * implies our reproduced results.

| Model | Method | N:M Pattern | Top-1 Accuracy (%) | Epochs (Train) | FLOPs (Train) |
|---|---|---|---|---|---|
| ResNet-18 | Baseline | - | 70.9 | 120 | $1\times$(1.4e18) |
| ResNet-18 | ASP | 2:4 | 69.9 | 200 | 1.24$\times$ |
| ResNet-18 | SR-STE | 2:4 | 71.2 | 120 | 0.83$\times$ |
| ResNet-18 | **LBC** | 2:4 | **71.5** | 120 | **0.70$\times$** |
| ResNet-50 | Baseline | - | 77.1 | 120 | $1\times$(3.2e18) |
| ResNet-50 | ASP | 2:4 | 76.8 | 200 | 1.24$\times$ |
| ResNet-50 | SR-STE | 2:4 | 77.0 | 120 | 0.83$\times$ |
| ResNet-50 | **LBC** | 2:4 | **77.2** | 120 | **0.72$\times$** |
| ResNet-50 | SR-STE | 1:4 | 75.3 | 120 | 0.74$\times$ |
| ResNet-50 | **LBC** | 1:4 | **75.9** | 120 | **0.51$\times$** |
| ResNet-50 | SR-STE | 2:8 | 76.2 | 120 | 0.74$\times$ |
| ResNet-50 | **LBC** | 2:8 | **76.5** | 120 | **0.53$\times$** |
| ResNet-50 | SR-STE | 1:16 | 71.5 | 120 | 0.69$\times$ |
| ResNet-50 | **LBC** | 1:16 | **71.8** | 120 | **0.38$\times$** |
| DeiT-small | Baseline | - | 79.8 | 300 | $1\times$(8.9e18) |
| DeiT-small | SR-STE* | 2:4 | 75.7 | 300 | 0.83$\times$ |
| DeiT-small | **LBC** | 2:4 | **78.0** | 300 | **0.71$\times$** |

Tesla A100 GPUs. Following Zhou *et al.* [40], we train the ResNets for 120 epochs with an initial learning rate of 0, which is linearly increased to 0.1 during the first 5 epochs and then decayed to 0 scheduled by the cosine annealing. The SGD is adopted to update parameters with the weight decay and momentum setting as 0.0005 and 0.9. Besides, we adopt the Timm framework [38] to train DeiT with 300 epochs. To implement LBC, we set $t_i$ to 0 and $t_f$ to 1/2 of the total training epochs. Top-1 classification accuracy, training epochs and training FLOPs are reported.

In Sec. 4.2, we compare our LBC with existing N:M sparsity methods including ASP [33] and SR-STE [40] under different N:M patterns. In Sec. 4.3, we further show the performance of LBC and traditional unstructured sparsity methods including RigL [7], GMP [42], and STR [17] under similar total compression rates. We give performance analyses of LBC *w.r.t.* its components in Sec. 4.4.

## 4.2 Comparison with N:M sparsity methods

**Image classification**. We first apply our LBC to train N:M sparse ResNet [12] and DeiT [36] on ImageNet [4]. Tab. 1 shows that the proposed LBC takes the lead in all N:M patterns with even less training burden for sparsifying ResNet with depth of 18 and 50. For ResNet-18, LBC outperforms SR-STE [40] at 2:4 pattern at the Top-1 classification accuracy by 0.3% (71.2% for SR-STE and 71.5% for LBC). For ResNet-50, LBC again demonstrates its superiority against both ASP [30] and SR-STE for all sparse patterns. Specifically, ASP achieves a Top-1 accuracy of 76.8% for training 2:4 sparse ResNet-50 using 200 training epochs that contain pre-training and fine-tuning phase, while LBC obtains a significantly higher accuracy of 77.2% with a fewer epoch of 120. When it comes to 1:4 sparse case, LBC surpasses SR-STE [40] by a large margin (75.9% *v.s.* 75.3% in Top-1 accuracy and 0.49$\times$ *v.s.* 0.74$\times$ in training FLOPs). Although SR-STE also alleviates the pre-training process, the dense backward propagation greatly decreases its training efficiency in comparison with LBC. From Fig.5 (a), LBC is still advantageous in all aspects when compared with SR-STE in the 1:4 case.

We further conduct experiments on the recent advanced Vision Transformer (ViT) model DeiT [36] to validate the efficacy of LBC. Similar to the strategy for compressing CNNs, we divide all parameters including the multi-head attention and feed-forward module in DeiT-small and train the N:M sparse networks from scratch. Tab. 1 shows the results of LBC and our re-produced SR-STE for training 2:4 sparse DeiT on ImageNet. As can be seen, LBC again surpasses SR-STE by a noticeable margin of 2.3% (78.0% for LBC and 75.7% for SR-STE) using fewer training FLOPs (0.71$\times$ for LBC and 0.83$\times$ for SR-STE). Therefore, the ability of LBC is well validated for compressing ViT models.

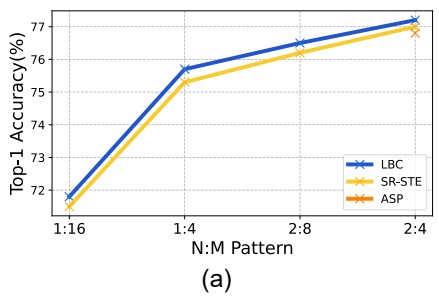 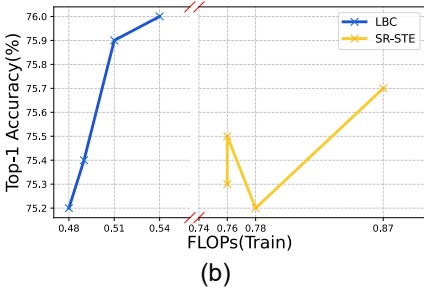

(a)                                            (b)

Figure 3: (a) Comparison results under different N:M pattern for sparsifying ResNet-50. (b) Top-1 accuracy *v.s.* training FLOPs for SR-STE and our proposed LBC at 1:4 case.

Table 2: Results on object detection.

| Model | Method | N:M Pattern | mAP. |
|-------|--------|-------------|------|
| F-RCNN | Baseline | - | 37.4 |
| F-RCNN | SR-STE | 2:4 | 38.2 |
| F-RCNN | **LBC** | 2:4 | **38.5** |
| F-RCNN | SR-STE | 2:8 | 37.2 |
| F-RCNN | **LBC** | 2:8 | **37.3** |

Table 3: Results on instance segmentation.

| Model | Method | N:M Pattern | Box mAP | Mask mAP |
|-------|--------|-------------|---------|----------|
| M-RCNN | Baseline | - | 38.2 | 34.7 |
| M-RCNN | SR-STE | 2:4 | 39.0 | 35.3 |
| M-RCNN | **LBC** | 2:4 | **39.3** | **35.4** |
| M-RCNN | SR-STE | 2:8 | 37.6 | 33.9 |
| M-RCNN | **LBC** | 2:8 | **37.8** | **34.0** |

**Object detection and instance segmentation**. Next, we exploit the efficacy of LBC on object detection and instance segmentation. Tab. 2 and Tab. 3 respectively display the comparison results of LBC against SR-STE [40] for training 2:4 sparse Faster-RCNN [8] and Mask-RCNN [11] with the backbone of ResNet-50 on the COCO benchmark [22]. The results further demonstrate the robustness and superiority of LBC on other computer vision tasks.

### 4.3 Comparison with unstructured sparsity methods

We also compare the performance of LBC with state-of-the-art unstructured sparsity methods in Tab. 4. By training 1:16 sparse ResNet-50 on ImageNet, LBC achieves a Top-1 accuracy of 71.8%, with 1.4% and 1.2% improvements over STR and GMP, respectively. Furthermore, the compressed models generated by unstructured sparsity remain an irregular format, which brings great challenges for the real-application deployment. Nevertheless, under a similar sparse rate, our LBC achieves practical compression and acceleration thanks to the NVIDIA Ampere Core while still bringing higher accuracy performance, which validates the advantages of exploring N:M sparsity.

Table 4: Results of the N:M and unstructured sparsity methods for sparsifying ResNet-50.

| Method | Top-1 Accuracy (%) | Sparsity (%) | Params (Test) | FLOPs (Test) | Structured |
|--------|--------------------|--------------|---------------|--------------|------------|
| Baseline | 77.3 | 0.0 | 25.6M | 4.10G | ✓ |
| DNW | 68.3 | 95 | 1.28M | 0.20G | ✗ |
| RigL | 70.0 | 95 | 1.28M | 0.49G | ✗ |
| GMP | 70.6 | 95 | 1.28M | 0.20G | ✗ |
| STR | 70.4 | 95 | 1.27M | 0.16G | ✗ |
| SR-STE | 71.5 | 94 (1:16) | 3.52M | 0.44G | ✓ |
| **LBC** | **71.8** | 94 (1:16) | 3.52M | 0.44G | ✓ |

### 4.4 Performance analyses

**Training efficiency.** In this part, we investigate the property of LBC for improving the N:M sparse training efficiency. All the experimental results are conducted on ImageNet by training 1:4 sparse ResNet-50. We first point out that the total training FLOPs of LBC are proportional to the initial epoch $t_i$ and final epoch $t_f$. To explain, larger $t_i$ and $t_f$ indicate a more smooth curve for removing combinations and more weights can be sufficiently trained before the epoch reaches $t_f$. For a fair

Table 5: Experimental results for SR-STE and LBC at the same gradual sparsity schedule.

| Method | $t_i$ | $t_f$ | Top-1 Accuracy (%) | FLOPs (Train) |
|--------|-------|-------|--------------------|---------------|
| SR-STE | 0 | 0 | 75.3 | 0.74× |
| LBC | 0 | 0 | 75.2 | 0.48× |
| SR-STE | 0 | 30 | 75.5 | 0.76× |
| LBC | 0 | 30 | 75.4 | 0.49× |
| SR-STE | 0 | 60 | 75.3 | 0.78× |
| LBC | 0 | 60 | 75.9 | 0.51× |
| SR-STE | 30 | 60 | 75.6 | 0.87× |
| LBC | 30 | 60 | 76.0 | 0.54× |

Table 6: Experimental results for different criteria to train N:M sparse ResNet-50.

| Method | N:M Pattern | Top-1 Accuracy (%) |
|--------|-------------|--------------------|
| Baseline | - | 77.3 |
| Magnitude | 2:4 | 75.8 |
| Gradient | 2:4 | 76.1 |
| **S**-inverse | 2:4 | 72.9 |
| **S** (LBC) | 2:4 | 77.2 |

comparison, we also apply the same gradual sparsity schedule to SR-STE [40]. As illustrated in Tab. 5, more training FLOPs consistently improve the performance of LBC as the importance score can be jointly trained with weights more sufficiently. Fig. 3(b) further shows that our LBC takes the lead in the trade-off between training cost and Top-1 accuracy performance when compared with SR-STE [30, 40]. It is worthwhile to note that, by selecting different $t_i$ and $t_f$, LBC can provide a satisfying solution between training expenditure and performance under different resource constraints. Thus, the efficiency and applicability of LBC are significant.

**Combination Score.** To investigate our combination score **S**, we introduce three other criteria including 1) inversely sorted learned score **S**, 2) the magnitude-based criterion [10], and 3) the gradient-based criterion [28] under the same experiment settings for sparsifying ResNet-50 on ImageNet. Results in Tab. 6 show that the proposed learnable score significantly outranks other criteria. Thus, the efficacy of LBC for recommending best weight combinations is well demonstrated.

## 5 Limitation

Firstly, LBC still needs relatively dense computation in the early stage of N:M training as there are still plenty of left candidate subsets. Thus, despite the advantages over approaches that rely on pre-trained models or dense-gradient calculation over the entire training phase, there are still room for improving the efficiency of LBC. Secondly, this paper only explores N:M sparsity on computer vision tasks. We expect to show more results on other tasks such as natural language processing (NLP) in our future work. Finally, we mainly focus on N:M sparsity on weights, while adding N:M sparsity for gradient [15] and activation [2] in the workflow of LBC remains to be excavated in the near future.

## 6 Conclusion

N:M fine-grained sparsity is an important technique that allows fast inference and training on the next-generation hardware and platforms. In this paper, we have presented LBC towards efficient N:M sparsity from scratch. We first transfer the N:M learning into a combinatorial problem, which motivates us to select the best weights combination from $C_M^N$ candidates. Further, we propose to assign learnable importance scores for each combination, which is jointly optimized with the associate weights. We perform an in-depth analysis that the best combination can be identified by aggressively removing less important combinations. Extensive experiments have demonstrated the efficacy of LBC in reducing the N:M training cost, and its superiority over several SOTAs. Our LBC is characterized by end-to-end implementation, efficient training, the tractability of optimization, and state-of-the-art performance in sparsifying modern networks.

## Acknowledgement

This work is supported by the National Science Fund for Distinguished Young (No.62025603), the National Natural Science Foundation of China (No.62025603, No. U1705262, No. 62072386, No. 62072387, No. 62072389, No, 62002305, No.61772443, No. 61802324 and No. 61702136) and Guangdong Basic and Applied Basic Research Foundation (No.2019B1515120049).

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
