# OpenReview forum: "Learning Best Combination for Efficient N:M Sparsity"
_NeurIPS.cc/2022/Conference — NeurIPS 2022 Accept_

### Official Review · Reviewer_biEJ · 2022-07-11

**Rating:** 6
**Confidence:** 3
**Soundness:** 3 good
**Presentation:** 3 good
**Contribution:** 3 good

**Summary:**

This paper deals with the problem of N:M sparsity and proposes an effective method with structure parameter to weigh the importance of different combination. The final result is appealing.

**Questions:**

The problem of training cost saving should be corrected.

**Ethics Review Area:**

["I don’t know"]

**Limitations:**

The problem of training cost saving should be corrected.

**Strengths And Weaknesses:**

Strengths:
1. Good final pruning results


Weakness:
1. From the analysis of [1] Efficient Neural Network Training via Forward and Backward Propagation Sparsification, the gradient-based sparse training method cannot exceed the accelearation above a upper bound of around 1.5x since the gradient to structure parameters require dense gradient calculation, even though the gradient to weights is sparsified and the forward propagation is sparsified. I think this is a fundamental problem and unless the discussions around training costs are corrected, it should not be published. Otherwise it will again mislead the community. If this issue can be corrected, I am happy to raise the rating score to be positive.

After rebuttal: I got the difference. The gradient calculation to auxiliary parameters only finally cares about the gradient to weights. I raise the score.

---

> ### Author Response · Authors · 2022-07-29
> **Response to Reviewer biEJ**
>
> Thanks for your kind comments. We sincerely wish our response can well address your concerns so as to raise the rating score. We believe we introduce a sound approach and have made a strong contribution to N:M sparsity, which are also highly-appraised by Reviewer VQRc and Reviewer 378D.
>
> **Q1**: *From the analysis of [1] Efficient Neural Network Training via Forward and Backward Propagation Sparsification, the gradient-based sparse training method cannot exceed the acceleration above an upper bound of around 1.5x since the gradient to structure parameters require dense gradient calculation, even though the gradient to weights is sparsified and the forward propagation is sparsified. I think this is a fundamental problem and unless the discussions around training costs are corrected, it should not be published. Otherwise it will again mislead the community. If this issue can be corrected, I am happy to raise the rating score to be positive.*
>
> **A1**: We deeply understand your concern about the sparsity acceleration, which indeed motivates us to focus on N:M sparsity. Specifically, as illustrated in [1], the gradients of the pruned weights are non-zero as the pruned weights still participate in the forward propagation albeit their zero values. Therefore, the dense backpropagation is unavoidable in existing sparse training methods.
>
> Luckily, this obstacle in traditional irregular network sparsity has been well addressed under the setting of N:M regular sparsity. The unique characteristics in N:M has been well supported by NVIDIA Ampere Core to reach sparse training.
> Please kindly refer to figure 12, page 32 in [2] (link: https://www.nvidia.com/content/dam/en-zz/Solutions/Data-Center/nvidia-ampere-architecture-whitepaper.pdf). As can be seen, the A100 Sparse Tensor Core **skip** the compute of pruned weights instead of treating them as 0s, leading to a smaller matrix multiply and achieving a 2x speedup in 2:4 sparsity. Therefore, **the pruned weights no longer participate into both forward and backward propagation.** Indeed, the upper bound for reducing the training cost of N:M sparse training is (M/N)x.
>
> Thanks again for your constructive comments on our work. We hope this rebuttal does well address your concerns. Besides, we will add a discussion between N:M sparsity and the claim in [1] in our final version to avoid any possible confusion for the community. If you have any other questions, please let us know and we are more than glad to discuss with you.
>
> [1] Efficient Neural Network Training via Forward and Backward Propagation Sparsification. In NeurIPS, 2021.
>
> [2] Nvidia a100 tensor core gpu architecture. https://www.nvidia.com/content/dam/en-zz/Solutions/Data-Center/nvidia-ampere-architecture-whitepaper.pdf, 2020.

---

> ### Author Response · Authors · 2022-08-08
> **Reminder**
>
> Dear Reviewer biEJ,
>
> Thanks again for your time and efforts. As the deadline for discussion is approaching, it would be nice of you to let us know whether our answers have solved your concerns so that we can better improve our work. We are happy to provide any additional clarifications that you may need.
>
> Best wishes,
>
> Paper1092 Authors

---

### Official Review · Reviewer_378D · 2022-07-11

**Rating:** 6
**Confidence:** 4
**Soundness:** 2 fair
**Presentation:** 3 good
**Contribution:** 3 good

**Summary:**

Authors tackle the problem of a learnable structural sparsity parametrized by N:M pattern. The best sparsity patterns per layer are found via divide-and conquer like algorithm where from a large set of proposals only one is selected in the end. At first, all possible candidate combinations are proposed, then, during training they are ranked according to the importance variable. At every algorithm step, the combination with the lowest score is removed from the pool. Finally, the last standing combination is selected. Experiments are performed on Resnet50 and DIET model on Imagenet.


**Questions:**

Formulation of the implementation is a bit confusing. Does it mean that during training all combinations contribute to the output of the layer? If true it will mean higher memory cost during training and higher computational complexity.
From another perspective, if S is implemented via straight through estimator than it is sampled during the training and the gradient propagates only to the elements with S = 1. Please clarify this point.
Score matrix. Does it mean that the score is a real value learned by the model? Is there any evidence that it does reflect importance? Authors provide derivation to end up with magnitude times gradient equation. However, providing some empirical evidence will be beneficial.
Can authors explain how to set newly introduced hyper-parameters and what is the intuition behind the choice?
Was the training recipe exactly the same when compared to SR-STE?

**Ethics Review Area:**

["I don’t know"]

**Limitations:**

At my current understanding, the cost of initial epochs is going to be higher due the requirement of forward0backward to be performed over all candidates. It will be nice to have some latency study as authors primary focused on efficiency of N:M sparsity.

**Strengths And Weaknesses:**

Paper provide a comprehensive overview of existing pruning paradigms (structured and unstructured). Ampere sparsity N:M is also clearly introduced.
The problem of structural pruning is usually not formulated a combinatorial problem. The fact that authors do makes the paper stronger and the approach sound. Particularly, formulation in section 3.2 is sufficient. The implementation of learning score matrix S is a bit unclear and answering questions from the section below will help to understand the paper better.
The code is available to reviewers, I went over it but didn't run.

---

> ### Author Response · Authors · 2022-07-29
> **Response to Reviewer 378D**
>
> Many thanks for your responsible reviews that will help us improve the manuscript. Please see the following answers to your questions.
>
> **Q1**: *Does it mean that during training all combinations contribute to the output of the layer? If true it will mean higher memory cost during training and higher computational complexity. From another perspective, if S is implemented via straight through estimator than it is sampled during the training and the gradient propagates only to the elements with S = 1.*
>
> **A1**: First, it is true that all combinations contribute to the output since the combination scores participate in the forward and backward propagation. However, please note that it leads to negligible costs on computation and memory since only one binary mask is finally introduced to indicate the removal/preservation of weights without any complex matrix multiplication (see Eq.(8)). In the backpropagation, the gradient of combination scores is obtained by a simple dot multiplication as W*∂L/∂W. Therefore, the overall memory cost as well as computational complexity are still drastically reduced comparing to other methods with dense gradient calculation [1] and pre-training burden [2], which are avoidable in our method (see **Q2** for an experimental validation).
>
> [1] Learning N: M fine-grained structured sparse neural networks from scratch. In ICLR, 2020
>
> [2] Nvidia a100 tensor core gpu architecture. https://www.nvidia.com/content/dam/en-zz/Solutions/Data-Center/nvidia-ampere-architecture-whitepaper.pdf, 2020.
>
> **Q2**: *At my current understanding, the cost of initial epochs is going to be higher due the requirement of forward \& backward to be performed over all candidates. It will be nice to have some latency study as authors primary focused on efficiency of N:M sparsity.*
>
> **A2**: Please kindly refer to our answer to **Q1** first. Following your valuable suggestion, we provide the training time comparison for sparsifying ResNet-50 on ImageNet below.
>
> | N:M   Pattern | Method | Training Time (NVIDIA A100 GPU days) |
> | ------------- | ------ | -------------------------------------- |
> |      2:4         |  Baseline    |                3.32                        |
> |      2:4         |   ASP     |                       4.12                 |
> |      2:4         |   SR-STE     |                        2.76                |
> |      2:4         |   LBC     |                               2.39        |
>
> As can be can, our LBC can effectively reduce the training latency for N:M sparse training. These results will be included in our final version.
>
> **Q3**: *Score matrix. Does it mean that the score is a real value learned by the model? Is there any evidence that it does reflect importance? Authors provide derivation to end up with magnitude times gradient equation. However, providing some empirical evidence will be beneficial.*
>
> **A3**: Exactly, the score is indeed a real and learnable value. In Sec. 3.4, we have formally proved that our introduced score matrix can be used to measure the relative importance between candidate subsets.
>
> Following your constructive advice, we further introduce three additional criteria for comparison with our combination score. These three additional criteria include 1) inversely sorting our learned scores; 2) pruning using the magnitude-based criterion, 3) the gradient-based criterion. The experiments are performed using ResNet-50 on ImageNet and the results below well demonstrate the efficacy of the proposed learnable score for reflecting the importance of corresponding combinations.
>
> | Method                      | N:M Pattern | Top-1 Accuracy |
> | --------------------------- | ----------- | -------------- |
> | Combination score           |     2:4        |        77.2        |
> | Combination score (Inverse) |     2:4        |       72.9         |
> | Magnitude-based             |       2:4      |        75.8        |
> | Gradient-based              |        2:4     |          76.1      |
>
> **Q4**: *Can authors explain how to set newly introduced hyper-parameters and what is the intuition behind the choice?*
>
> **A4**: The introduced hyper-parameters in this paper include $t_i$ and $t_f$, respectively denoting the initial and final epochs to remove combinations. As analyzed in Sec 4.4, larger values of $t_i$ and $t_f$ enable more robust performance but also more training FLOPs. Therefore, the users can flexibly switch the values of these two hyper-parameters to achieve a trade-off between training costs and model performance according to the availability of hardware resources.
>
> **Q5**: *Was the training recipe exactly the same when compared to SR-STE?*
>
> **A5**: Definitely. For fair comparison, we keep the same training recipe with SR-STE including learning rate, training epoch, *etc*.

---

> ### Author Response · Authors · 2022-08-08
> **Reminder**
>
> Dear Reviewer 378D,
>
> Thanks again for your time and efforts. As the deadline for discussion is approaching, it would be nice of you to let us know whether our answers have solved your concerns so that we can better improve our work. We are happy to provide any additional clarifications that you may need.
>
> Best wishes,
>
> Paper1092 Authors

---

### Official Review · Reviewer_VQRc · 2022-07-15

**Rating:** 7
**Confidence:** 5
**Soundness:** 3 good
**Presentation:** 3 good
**Contribution:** 3 good

**Summary:**

This paper formulates the N:M sparsity as the combinatorial optimization problem.
This formulation is a well-motivated and promising direction. Then the authors use a learnable score matrix to measure
the candidate's importance. The authors use the gradients of group elements to update the score matrix.

**Questions:**

See Strengths And Weaknesses


###############################################

Post-rebuttal: I have read the rebuttal carefully, and all of my comments/questions were addressed. Thanks.

**Strengths And Weaknesses:**

Strengths:


This paper is well-written and easy to follow.

The performance gain is consistent on various sparse ratios and models.

At present, reducing the sparse model training cost can motivate the research community to study sparse training acceleration.

The source code is available.

Weaknesses:
The authors are encouraged to conduct experiments on other tasks.

The authors only report the training  FLOPs, the authors are encouraged to report the training time.

---

> ### Author Response · Authors · 2022-07-29
> **Response to Reviewer VQRc**
>
> Thanks for your constructive and supportive comments.
>
> **Q1**: *The authors are encouraged to conduct experiments on other tasks.*
>
> **A1**: Following your advice, we further conduct experiments on two other tasks including object detection and instance segmentation. Below displays the experimental results in which the proposed LBC performs best on both tasks.
>
> Object detection results on COCO:
>
> | Model      | Method | Sparse Pattern | mAP  |
> | ---------- | ------ | -------------- | ---- |
> | F-RCNN-R50 | -      | Dense          | 37.4 |
> | F-RCNN-R50 | SR-STE | 2:4            | 38.2 |
> | F-RCNN-R50 | LBC    | 2:4            |    38.5  |
> | F-RCNN-R50 | SR-STE | 2:8            | 37.2 |
> | F-RCNN-R50 | LBC    | 2:8            |   37.3   |
>
>  Instance segmentation results on COCO:
>
> | Model      | Method | Sparse Pattern | Box mAP | Mask mAP |
> | ---------- | ------ | -------------- | ------- | -------- |
> | M-RCNN-R50 | -      | Dense          | 38.2    | 34.7     |
> | M-RCNN-R50 | SR-STE | 2:4            | 39.0    | 35.3     |
> | M-RCNN-R50 | LBC    | 2:4            |     39.3    |   35.4       |
> | M-RCNN-R50 | SR-STE | 2:8            | 37.6    | 33.9    |
> | M-RCNN-R50 | LBC    | 2:8            |     37.8    |    34.0      |
>
> **Q2**: *The authors only report the training FLOPs, the authors are encouraged to report the training time.*
>
> The training time comparison for sparsifying ResNet-50 on ImageNet is provided in the following, which will also be included in our final version.
>
> | N:M   Pattern | Method | Training Time (NVIDIA A100 GPU days) |
> | ------------- | ------ | -------------------------------------- |
> |      2:4         |  Baseline    |                3.32                        |
> |      2:4         |   ASP     |                       4.12                 |
> |      2:4         |   SR-STE     |                        2.76                |
> |      2:4         |   LBC     |                               2.39        |

---

> ### Author Response · Authors · 2022-08-08
> **Reminder**
>
> Dear Reviewer VQRc,
>
> Thanks again for your time and efforts. As the deadline for discussion is approaching, it would be nice of you to let us know whether our answers have solved your concerns so that we can better improve our work. We are happy to provide any additional clarifications that you may need.
>
> Best wishes,
>
> Paper1092 Authors

---

> > ### Comment · Reviewer_VQRc · 2022-08-08
> > **After rebuttal**
> >
> > Thanks for your response, I have updated the official comments.

---

### Official Review · Reviewer_11EU · 2022-07-15

**Rating:** 4
**Confidence:** 5
**Soundness:** 2 fair
**Presentation:** 3 good
**Contribution:** 2 fair

**Summary:**

This work is focused on efficient learning of N:M sparsity from scratch. Finding the optimal N:M sparsity pattern is characterized as jointly solving a series of combinatorial problems with finite collections of candidates. The authors proposed to associate a learnable score parameter with each possible combination and learn it from data with the help of straight-through-estimator (STE). The experiments showed the proposed training method yields slightly better results with much fewer FLOPs compared to stronger baselines and much better performance than others.

**Questions:**

Please see the weaknesses part and address my questions there.

The superscript of $\mathcal{L}^l$ in the last term of eq. (9) seems to be in the wrong place.

**Limitations:**

Please see the weaknesses part and address my questions there.

**Strengths And Weaknesses:**

Strengths:

+ This work successfully showed that N:M sparsity under a kind of gradual pruning manner can work well on large modern neural networks. The gradual pruning enables the possibility of a efficient method that trains N:M sparse models from scratch so that the expensive dense training process can be avoided.

+ Experiments are promising in terms of performance and FLOPs.

--------------------------------

Weaknesses:

- The biggest issue of this work if over-claiming. Characterizing the problem as jointly solving a series of combinatorial problems is trivial, and more importantly, does not simply the original problem in any sense.  The authors claimed that they adopted a "divide-and-conquer" method. But according to my understanding of "divide-and-conquer", this claim is not true. The problem is not simplified into simpler forms with lower complexity. It is still a complex joint optimization problem. And the authors uses a simple method that learns a score for each combination. This type of methods is not new and can date back to network slimming (Liu et al., 2017).
- The only place that the learnable scores are used is eq. (8). But if I understand correctly, removing the S score in eq. (8) does not influence the whole methods at all. If that is the case, I am quite doubtful of what scores are learnt finally.
- For a paper that emphasizes its training efficiency, there is no energy or wall-time comparison of the proposed method and other baselines. This is related to another of my question that --- whether Nvidia A series GPU can exploit the gradually increasing N:M sparsity to improve training efficiency?

---

> ### Author Response · Authors · 2022-07-29
> **Response to Reviewer 11EU**
>
> Thanks for your in-depth review that will help us strengthen the manuscript. We hope our response can address your concern here.
>
> **Q1**: *Characterizing the problem as jointly solving a series of combinatorial problems is trivial, and more importantly, does not simply the original problem in any sense. The authors claimed that they adopted a "divide-and-conquer" method. But according to my understanding of "divide-and-conquer", this claim is not true. The problem is not simplified into simpler forms with lower complexity. It is still a complex joint optimization problem*.
>
> **A1**: With full respect, what we aim to deliver may be misunderstood. We do not intend to introduce traditional "divide-and-conquer" methods to solve our optimization problem. Here, our divide-and-conquer is much like algorithm where from a large set of proposals only one is selected in the end (**see summary of Reviewer 378D**). What we aim to deliver seems to be well understood by other reviewers. Besides, characterizing the problem as jointly solving a series of combinatorial problems is also supported or even highly appraised by other reviewers. We apologize if our word usage causes any confusion and wish that our explanation can well address your concern.
>
> **Q2**: *The authors uses a simple method that learns a score for each combination. This type of methods is not new and can date back to network slimming (Liu et al., 2017).*
>
> **A2**: With all due respect, though the concept of learnable scores exists in other methods, we would like to highlight that our LBC has its unique designs to solve N:M sparsity. We do not learn the importance of each weight as other methods. Instead, our learnable score is particularly designed to reflect the importance of each combination based on our reformulation of N:M sparsity, which has also been well proofed in Sec.3.4 of our paper. As a result, we achieve state-of-the-art performance compared with other N:M methods even with far fewer training costs.
>
> **Q3**: *The only place that the learnable scores are used is eq. (8). But if I understand correctly, removing the S score in eq. (8) does not influence the whole methods at all. If that is the case, I am quite doubtful of what scores are learnt finally.*
>
> **A3**: Nice question. Intuitively, removing S does not influence the result of Eq. (8) at least in the forward propagation. However, the motive of retaining S in Eq. (8) is actually to derive the gradient of S in the backward propagation (see Eq.(9)). Recall that in each training epoch, we remove a part of low-scored candidates (Lines 7-8, Alg.1) and then the binary mask is generated (Eq. (8)). If removed from Eq. (8), S will be frozen and unlearnable since it does not involve in the computing graph. Consequently, the generated binary mask is only related to the initialization of S. Therefore, S needs to be kept in Eq.(8). In Sec.3.4 of our paper, we have formally proved that the learned score S in this manner can well reflect the relative importance between different candidate subsets. Also, we further conduct experiments to show that our learnable score performs much better than existing criteria (**kindly refer to our answer to Q3 of Reviewer 378D**). To avoid confusion, the above discussion will be added in our paper (right after Eq.(8)).
>
> **Q4**: *For a paper that emphasizes its training efficiency, there is no energy or wall-time comparison of the proposed method and other baselines.*
>
> **A4**: We appreciate this valuable comment, which helps us to further improve the quality of our paper. The wall-time comparison for ResNet-50  at 2:4 pattern is provided below, where the proposed LBC shows a superior training efficiency compared with other baselines. This experiment will be included in our paper.
>
> | N:M   Pattern | Method | Training Time (NVIDIA A100 GPU days) |
> | ------------- | ------ | -------------------------------------- |
> |      2:4         |  Baseline    |                3.32                        |
> |      2:4         |   ASP     |                       4.12                 |
> |      2:4         |   SR-STE     |                        2.76                |
> |      2:4         |   LBC     |                               2.39        |
>
> **Q5**: *Whether Nvidia A series GPU can exploit the gradually increasing N:M sparsity to improve training efficiency?*
>
>  **A5**: Definitely. During the gradual pruning procedure, increasing number of blocks will satisfy the n:m pattern due to the removal of redundant combinations, which therefore can be supported by NVIDIA A100 GPU to improve training efficiency.
>
> **Q6:** *The superscript of $L^l$ in the last term of eq. (9) seems to be in the wrong place.*
>
> **A6**: We highly appreciate your careful review. It is a typo and will be corrected. Besides, a more comprehensive checking will be made in our final version.

---

> > ### Comment · Reviewer_11EU · 2022-08-09
> > **Thanks to the authors for the response. My concerns were only partially addressed**
> >
> > I would like to thank the authors for the response. The additional results and explanation cleared my questions Q4-Q6. But I think my questions 1-3 were not quite addressed.
> >
> > Q1: According to the definition on Wikipedia at https://en.wikipedia.org/wiki/Divide-and-conquer_algorithm, "divide-and-conquer" is a well established and widely accepted term in computer science. And the proposed method is definitely not a divide-and-conquer method. Misusing such a term is not a practice and it is the authors' responsibility to eliminate such confusion by using correct terms. It is not the audience's fault that they "misunderstood" what the authors tried to "deliver".
> >
> > Q2: I still don't buy the argument. The authors' just repeated my argument and I still found the *technique* used is not novel.
> >
> > Q3: I am not sure what will be learned for something that can be safely removed from the formulation.

---

> > > ### Author Response · Authors · 2022-08-09
> > > **Response to Reviewer 11EU**
> > >
> > > We sincerely thank you for your timely and constructive feedback. Also, we appreciate your effort in reviewing this paper. We believe we introduce a sound approach and have made a strong contribution to N:M sparsity, which are already appraised by the other three reviewers. Our further responses are provided below and we wish this time we can satisfy all your concerns.
> > >
> > > Response to Q1:
> > > We understand your concern. In our final version, we will rewrite contents w.r.t. “divide-and-conquer” to eliminate any possible confusion. We wish that more focuses can be put on what we have done in this paper when making your final decision. Thanks.
> > >
> > > Response to Q2:
> > > Full of respect, we first highly suggest going through other reviewers’ comments. This paper is well favored by all other reviewers. On one hand, our key contribution falls into the reformulation of N:M sparsity and an efficient solution that drastically reduces the training cost of existing approaches with even higher performance, which is appreciated by all reviewers. On the other hand, our learnable scores do quite differ from existing studies such as the mentioned network slimming. See our response to Q3, which may provide a clearer example to show how our score evolves and help achieve N:M sparsity.
> > >
> > > Response to Q3:
> > > Let us make a toy example.
> > > An importance score vector is initialized as S = [1, 1, 1, 1, 1, 1].
> > > If removing S from Eq.(8), S will be fixed to its initial statement since it does not involve in the computing graph as analyzed in our last response. Consequently, we have no idea which combinations should be removed.
> > > On the contrary, keeping S in Eq. (8) enables the updating of S. Therefore, at the next iteration, S can be changed, for example, to state [0.1, 0.8, 1.5, 0.9, 1.2, 1.3]. And then, the combination candidate with the lowest importance score 0.1 is removed. In this manner, N:M sparsity can be efficiently achieved by gradually removing low-scored candidate subsets along the network training.

---

> ### Author Response · Authors · 2022-08-08
> **Reminder**
>
> Dear Reviewer 11EU,
>
> Thanks again for your time and efforts. As the deadline for discussion is approaching, it would be nice of you to let us know whether our answers have solved your concerns so that we can better improve our work. We are happy to provide any additional clarifications that you may need.
>
> Best wishes,
>
> Paper1092 Authors

---

### Meta-Review · Area_Chair_oBFK · 2022-08-25

**Recommendation:** Accept
**Confidence:** Certain

**Metareview:**

The paper presents a novel method on training N:M sparse-weight neural networks, which can be significantly accelerated by NVIDIA A100 GPUs. The optimal N:M pattern can be found via jointly solving a series of combinatorial problems with finite collections of candidates. Majority of the reviewers found the paper

The AC believes that the concern of 11EU can be solved by re-phrasing the descriptions but doesn't affect the effectiveness and novelty of this paper.

**Award:**

No

---

### Decision · Program_Chairs · 2022-09-14

Accept